

# Population genetic structure and hybrid zone analyses for species delimitation in the Japanese toad (*Bufo japonicus*)

Kazumi Fukutani[1], Masafumi Matsui[1] and Kanto Nishikawa[1,2]

[1] Graduate School of Human and Environmental Studies, Kyoto University, Kyoto, Japan
[2] Graduate School of Global Environmental Studies, Kyoto University, Kyoto, Japan

## ABSTRACT

Hybridization following secondary contact may produce different outcomes depending on the extent to which genetic diversity and reproductive barriers have accumulated during isolation. The Japanese toad, *Bufo japonicus*, is distributed on the main islands of Japan. In the present study, we applied multiplexed inter-simple sequence repeat genotyping by sequencing to achieve the fine-scale resolution of the genetic cluster in *B. j. japonicus* and *B. j. formosus*. We also elucidated hybridization patterns and gene flow degrees across contact zones between the clusters identified. Using SNP data, we found four genetic clusters in *B. j. japonicus* and *B. j. formosus* and three contact zones of the cluster pairs among these four clusters. The two oldest diverged lineages, *B. j. japonicus* and *B. j. formosus*, formed a narrow contact zone consistent with species distinctiveness. Therefore, we recommend that these two subspecies be elevated to the species level. In contrast, the less diverged pairs of two clusters in *B. j. japonicus* and *B. j. formosus*, respectively, admixed over a hundred kilometers, suggesting that they have not yet developed strong reproductive isolation and need to be treated as conspecifics. These results will contribute to resolving taxonomic confusion in Japanese toads.

## INTRODUCTION

Hybrid zones are natural laboratories that offer many insights into speciation processes, thereby contributing to a more detailed understanding of evolution (*Barton & Hewitt, 1985*; *Hewitt, 1988*; *Abbott et al., 2013*). Hybridization following secondary contact may produce different outcomes depending on the extent to which genetic diversity and reproductive barriers have accumulated during isolation. This results in a reduction in differentiation as well as the fusion of gene pools. Alternatively, an increase in the strength of the genetic barrier may lead to complete reproductive isolation (*Barton & Hewitt, 1985*; *Wu, 2001*).

Hybridization is frequent and evolutionarily significant in amphibians (*Burbrink & Ruane, 2021*). There are well-described studies on amphibians for the hybrid zone of fire-bellied toads (*Bombina bombina* and *B. variegata*; *e.g.*, *Szymura & Barton, 1986*, *1991*; *Yanchukov et al., 2006*), green toads (*Bufotes viridis* subgroups; *e.g.*, *Stöck et al., 2006*;

Corresponding author
Kazumi Fukutani,
fukutani.kazumi.55a@kyoto-u.jp

*Colliard et al., 2010*; *Dufresnes et al., 2014*), and, more recently, European common toads (*Bufo bufo* and *B. spinosus*; *e.g.*, *Arntzen et al., 2016*; *Dufresnes et al., 2020a*; *van Riemsdijk et al., 2023*). In contrast to many other anuran species, the hybrid zone of Japanese toads has not yet been examined in detail (*Bufo japonicus* subspecies; *Miura, 1995*). However, they have the advantage of comprising distinct genetic lineages representing different stages of the speciation process because several contact zones of the different genetic lineages have been recognized (*Fukutani et al., 2022*). Regarding amphibian cases, the extent of natural hybridization in contact zones has been correlated with divergence times (*Hickerson, Meyer & Moritz, 2006*; *Dufresnes et al., 2021*).

*Bufo japonicus* Temminck and Schlegel, 1838 is widely distributed in the Japanese archipelago, Honshu, Shikoku, Kyushu, and some adjacent islands. This species is divided into two subspecies, *B. j. japonicus* from western Japan and *B. j. formosus* *Boulenger, 1883* from eastern Japan. These two subspecies are parapatrically distributed with the boundary in the Kinki region of central Japan (*Matsui & Maeda, 2018*). *Matsui (1984)* concluded that *B. j. japonicus* and *B. j. formosus* showed a climatic cline in their morphometric characteristics, which was insufficient to distinguish them as different species because of their identity in the fundamental patterns of modes of life. However, *Dufresnes & Litvinchuk (2021)* recently proposed elevating *B. j. japonicus* and *B. j. formosus* to the species level based on the Miocene split estimated by mtDNA markers. However, they refrained from taxonomic changes because mitochondrial distances may not reflect actual species distances. Other studies proposed the Kinki region as a hybrid zone of *B. j. japonicus* and *B. j. formosus* by a C-banding analysis of chromosomes (*Miura, 1995*).

The sympatric distribution of mitochondrial haplotypes of *B. j. japonicus* and *B. j. formosus* was also found in the Kinki region (*Fukutani et al., 2022*). Furthermore, several contact distributions of the genetic lineages in the two subspecies were identified. These findings indicate the necessity of analyzing the degree of hybridization between the two subspecies and other genetic lineages for taxonomic revision.

The delimitation of species must be connected to a species concept. We used the integrative species concept (*Queiroz, 2007*, *2020*) that considers both aspects, phylogeny and the reproductive isolation mechanism.

In this study, we applied multiplexed ISSR genotyping by sequencing (MIG-seq: *Suyama & Matsuki, 2015*) to achieve the fine-scale resolution of genetic clusters in *B. j. japonicus* and *B. j. formosus*. MIG-seq has been effectively used to study molecular phylogenetic taxonomy for various taxa (see *Suyama et al., 2022*).

We performed cline analyses to elucidate the degree of gene flows. The results of cline analyses explained the transition between the characteristics of interbreeding species across the hybrid zone and will contribute to a more detailed understanding of the mechanisms maintaining species boundaries (*Barton & Hewitt, 1985*). Valid species need to exhibit significant divergence and narrow transition zones. In contrast, insufficiently diverged lineages that remained conspecific need to admix freely across broad genetic areas. We revised the taxonomic status of *B. j. japonicus* and *B. f. formosus* based on phylogenetic and hybrid zone analyses.

## MATERIALS AND METHODS

### Sampling and MIG-seq

A total of 155 individuals of *B. japonicus* and 13 of *B. torrenticola* Matsui, 1976 were collected, covering the complete distribution range (Table S1). The Animal Experimentation Ethics Committee in the Graduate School of Human and Environmental Studies, Kyoto University approved this research (20-A-5, 20-A-7, 22-A-2). DNA was extracted from frozen or ethanol-preserved tissue samples (*e.g.*, muscle, liver, or skin) with the Qiagen DNeasy Blood and Tissue Kit following the manufacturer's instructions.

We prepared two genomic libraries and sequenced them separately for the convenience of the experiment, and the data obtained were analyzed together as described below. Library 1 included 121 DNA samples of *B. japonicus* and 13 of *B. torrenticola*, while library 2 had 40 DNA samples of *B. japonicus*, with six of *B. japonicus* overlapping in both libraries (Table S1). The two genomic libraries were prepared following the protocol described by *Matsui et al. (2019)* for library 1 and that described by *Watanabe et al. (2020)* for library 2. Amplicons in libraries 1 and 2 were purified and sequenced on the Illumina MiSeq Sequencer (Illumina, San Diego, CA, USA) using the MiSeq Reagent Kit v3 (150 cycles; Illumina, San Diego, CA, USA). Two libraries were prepared and sequenced separately for the convenience of the molecular experiment, and the raw sequence data obtained were combined for subsequent data analyses.

The raw sequence reads of MIG-seq data were deposited in the DNA Data Bank of Japan (DDBJ) Sequence Read Archive (DRA) under accession number DRA016475 (BioProject ID; PRJDB15971: BioSample ID; SAMD00622809–SAMD00622982).

Raw paired-end sequences (reads 1 and 2) were filtered by fastp version 0.23.2 (*Chen et al., 2018*) to trim the first 14 base sequences of read 2 and the primer regions of reads 1 and 2 and to discard reads shorter than 80 bp and low-quality sequences with phred quality Q < 30 according to *Suyama & Matsuki (2015)*. We then mapped the filtered reads to reference sequences because mapping obtains more loci than a *de novo* analysis of MIG-seq data (*Takata et al., 2021*). As the reference genome sequence for Japanese toads, we used the genome assembly of their closely related species, *B. gargarizans* (RefSeq assembly accession number: GCF_014858855.1; https://www.ncbi.nlm.nih.gov/data-hub/genome/GCF_014858855.1/). The assembly contained 11 chromosome-level contigs and unplaced scaffolds. We ultimately mapped the filtered reads to the indexed reference sequences by bwa-mem2 version 2.2.1 (*Vasimuddin et al., 2019*) to make SAMfiles, which were then converted to BAM files and sorted with a minimum mapping quality of 20 using *samtools* version 1.15 (*Li et al., 2009*).

### Genotyping

We prepared the following datasets: dataset I, data from samples of *B. japonicus* and *B. torrenticola* to examine the genetic structure of Japanese toads, and dataset II, data from samples of *B. j. japonicus* and *B. j. formosus* to investigate the degree of reproductive isolation between the two subspecies. We excluded the 11 samples from these two datasets that were considered to be from artificially introduced populations based on a previous

study (*Fukutani et al., 2022*). We instead prepared dataset III, which included these 11 samples with dataset II to verify their genetic assignment in the population.

The reference-based analysis pipeline with the *gstacks* program followed by the *populations* program in Stacks v2.60 (*Rochette, Rivera-Colón & Catchen, 2019*) was applied to the mapped reads of all datasets to call SNPs and genotypes. The following filters were used for the *populations* program in Stacks. We initially kept variant sites with a minimum allele count of three (–min-mac 3) to ensure that an allele was in at least two diploid samples (*Rochette, Rivera-Colón & Catchen, 2019*). We then set up the maximum observed heterozygosity at 0.5 (–max-obs-het 0.50) because heterozygosity for a biallelic SNP was expected to be <0.5, and SNPs with values above this threshold may belong to paralogous loci or multilocus contigs (*Hohenlohe et al., 2011*; *Willis et al., 2017*). Subsequently, only one random SNP per locus was extracted (–write-random-snp) to avoid any effect of linkages among SNPs on the multivariate analysis (*Gargiulo, Kull & Fay, 2021*). In the population designation in a *population map*, we set two populations corresponding to *B. japonicus* and *B. torrenticola* for dataset I. In datasets II and III, we set two populations based on the admixture proportion ($q$-value, with $q$-value = 0.5 as a boundary) at the optimal number of clusters ($K$) = 2 in the Structure analysis (see below: *Pritchard, Stephens & Donnelly, 2000*) of dataset I. We ultimately only processed the loci present in at least 80% of samples in a population (-$r$ = 0.80) and those present in two populations for all datasets (-$p$ = 2). In the following stacks program, the two parameters, -$r$ and -$p$, varied, and the others were common for each analysis.

## Estimation of genetic structures

To estimate the population genetic structures of *B. japonicus* and *B. torrenticola*, we performed three different methods using SNP genotyping information and compared grouping among these methods: a discriminate analysis of principal components (DAPC; *Jombart, Devillard & Balloux, 2010*), Structure 2.3.4 (*Pritchard, Stephens & Donnelly, 2000*), and a principal component analysis (PCA; *Cavalli-Sforza, 1966*). DAPC was used for the inference of the number of clusters. We used Structure analyses to perform a Bayesian clustering analysis. In addition, complementary to Structure analyses, we performed PCA.

DAPC was performed on dataset I in the R package *adegenet* 2.1.8 (*Jombart, 2008*; *Jombart, Devillard & Balloux, 2010*; *Jombart & Ahmed, 2011*). This method maximizes the variance among groups while minimizing variations within groups without making assumptions about the underlying population genetic model. This approach transforms multilocus genotype data using PCA to derive uncorrelated variables that are input for a discriminate analysis. The optimal groups were initially assessed using the *de novo* clustering method, *find.cluster*, testing $K$ values from 1 to 8, and the best $K$ value was selected with the Bayesian information criterion (BIC) method. This *de novo* clustering method and initial DAPC using the *dapc* function were run. The *optim.a.score* was then used to assess the optimal number of principal components (PCs) to retain. Once the optimal number of PCs was selected, a second DAPC analysis was conducted using this value.

The program Structure 2.3.4 (*Pritchard, Stephens & Donnelly, 2000*) performed the analysis by an admixture model with correlated allele frequencies based on the Bayesian clustering method to infer the population structure. Since excessive uneven sampling may increase bias on admixture proportions in the Structure analysis (*Toyama, Crochet & Leblois, 2020*), we reduced the sample size in Yakushima and Tanegashima from dataset I, called dataset I-2, and conducted Structure analyses. Structure analyses were performed for the number of clusters $K$ from 1 to 8, with ten runs for each $K$ value. Markov chain Monte Carlo (MCMC; *Metropolis et al., 1953*; *Hastings, 1970*) iterations of 100,000 were implemented for each run after an initial burn-in of 100,000. The parallelization of Structure 2.3.4 calculations was achieved using EasyParallel (*Zhao et al., 2020*) to reduce the computational time. The optimal number of clusters was inferred in StructureSelector (*Li & Liu, 2018*) with the Delta $K$ ($\Delta K$; *Evanno, Regnaut & Goudet, 2005*), MedMeaK, MaxMeaK, MedMedK, and MaxMedK (*Puechmaille, 2016*). StructureSelector integrated the CLUMPAK program (*Kopelman et al., 2015*) to cluster and merge data from independent runs and generate graphical representations of the results. In a Structure analysis, an admixed ancestry is modeled by assuming that an individual has inherited some proportion of its genome from its ancestors in the population (*Pritchard, Stephens & Donnelly, 2000*).

PCA was performed on dataset I using the R package *adegenet* 2.1.8 (*Jombart, 2008*; *Jombart & Ahmed, 2011*), and the first two eigenvectors were plotted in two dimensions.

Moreover, we conducted a Structure analysis of dataset III to identify the assignment of genomic clusters for samples from introduced populations, reducing the sample size in Yakushima and Tanegashima for the above reason as dataset III-2. A Structure analysis was performed on the number of clusters $K$ from 1 to 6, and other parameters were the same as above.

## Phylogenetic estimations

We used SNAPP 1.5.2 (*Bryant et al., 2012*) implemented in Beast v 2.6.7 (*Bouckaert et al., 2019*) to estimate phylogenetic relationships among population groups identified by our clustering. We selected four individuals for each population group and applied them to the stacks program (*-r* = 1.0 and *-p* = 5). We ran SNAPP for 10,000,000 iterations with mutation rates *u* and *v* = 1.0, a gamma distribution with *alpha* = 2 and *beta* = 200 for the lambda prior, and *alpha* = 1, *beta* = 250, *kappa* = 1, and *lambda* = 10 for *snapprior*, sampling every 1,000 steps. Convergence was examined using Tracer 1.7.2 (*Rambaut et al., 2018*), and the results obtained were visualized by Densitree 2.2.7 with a burn-in of 10%. The maximum clade credibility tree with posterior probability was calculated using TreeAnnotator version 2.6.7 (*Bouckaert et al., 2019*). To perform comparisons, we reconstructed a mitochondrial phylogenetic tree using the mitochondrial cytochrome *b* sequences from *Fukutani et al. (2022)* of the same individuals used to construct the SNP tree adding the sequence of *B. g. gargarizans* as the outgroup. RAxML version 8.2.12 (*Stamatakis, 2014*) was employed for 1,000 bootstrap iterations with the GTRGAMMA model to infer a maximum likelihood phylogenetic tree based on mitochondrial sequences.

## Effective estimates of migration surfaces

We visualized the spatial patterns of gene flow using Fast Estimation of Effective Migration Surfaces (FEEMS; *Marcus et al., 2021*) to assess the genomic context and geographic location of any historical barriers to migration in *B. japonicus*. FEEMS is an improvement of Estimated Effective Migration Surfaces (*Petkova, Novembre & Stephens, 2016*) and uses a Gaussian Markov Random Field model in a penalized likelihood framework. This method uses locality information and pairwise dissimilarity matrices calculated from SNP data to identify regions where genetic similarity decays more quickly than expected under isolation by distance (*Petkova, Novembre & Stephens, 2016*). To estimate effective migration parameters, we used the genotype data of dataset II as well as the coordinate information of each individual as inputs. A polygon grid was prepared using QGIS 3.28. Cross-validation was performed and the lambda with the lowest cross-validation value was used to generate the final plot.

## Hybrid zone analyses

To estimate the geographic gradient of genomic differences between adjacent clusters of *B. japonicus*, we calculated the steepness of the cline of genetic differences. Assuming similar dispersal abilities among the individuals of each cluster and no geographic barriers to gene flow at their transitions, wide hybrid zones will be present for the younger pairs if they did not yet evolve significant reproductive isolation. In contrast, narrow transitions will be present for the older pairs if they represent distinct species.

We fit clines to the Structure $q$-value across the geographic transition between genetic clusters using the R package *hzar* version 0.2-7 (*Derryberry et al., 2014*). The admixture proportions inferred by the Structure program (*Pritchard, Stephens & Donnelly, 2000*) have frequently been used to fit a geographic consensus cline, from which the width of the hybrid zone is estimated (*e.g.*, *Tominaga et al., 2018*; *Dufresnes et al., 2020b*). To avoid bias on the admixture proportions of Structure, we also reduced the sample size in Yakushima and Tanegashima from dataset II as dataset II-2. We also fit clines to mitochondrial haplogroup frequency data from our previous study (Table S1; *Fukutani et al., 2022*) for comparison with nuclear ancestry clines.

We performed the stacks program on this subset, setting four populations based on the results of DAPC on dataset II, with $-r = 0.80$ and $-p = 4$, and conducted a Structure analysis using the same parameters as above. This subset was divided into sub-datasets I, II, and III, based on the $q$-value at $K = 4$ with some samples overlapping. Each sub-dataset contained individuals of two pure clusters, considering a $q$-value > 0.90 as pure individuals and admixed individuals between pure clusters. We applied the stacks program for each sub-dataset, setting three populations (two pure and one admixed population) with $-r = 0.80$ and $-p = 3$, and conducted a Structure analysis. The $q$-values on $K = 2$ for each sub-dataset were used to perform *hzar*. In addition to the three sub-datasets, we prepared sub-dataset III-2, which is data excluding samples in Shikoku and Seto Inland Sea (see discussion) and performed a similar analysis to that for the other sub-datasets.

We reduced the two-dimensional space (latitude and longitude) into a single-dimensional distance from the center line of the hybrid zone. The probable center

line of the admixture was estimated using R package *tess3r* version 1.1 (*Caye et al., 2016*, *2018*) and considered to be the baseline for *hzar*. The minimum distances from the baseline to individuals were calculated in QGIS 3.28. We assigned a positive or negative sign to these distances depending on individual orientations to the baseline.

The shape of a cline is modeled by combining three equations (*Szymura & Barton, 1986*, *1991*) that describe a sigmoid shape at the center of a cline (maximum slope) and two exponential decay curves on either side of the central cline (tails). We tested 15 different models, which combined three trait intervals and five fitting tails, for each cline plus a null model with no cline. The three possible combinations of trait intervals were used to scale clines by the minimum ($p_{min}$) and maximum ($p_{max}$) values in the cline: no scale (fixed to $p_{min} = 0$ and $p_{max} = 1$), observed values (fixed to $p_{min} = $ minimum observed mean values, $p_{max} = $ maximum observed mean values), and estimated values ($p_{min}$ and $p_{max}$ as the free parameter). The five possible combinations of fitting tails represent the cline shapes: no tails, right tail only, left tail only, symmetrical tails, mirror tails, and both tails estimated separately.

MCMC was performed for each model with the default values of 100,000 generations, each with a randomly selected seed and 10% of steps discarded as a burn-in. After each run, we compared the model performance using the Akaike information criterion score corrected for a small sample size (AICc; *Anderson & Burnham, 2002*). The model with the lowest AICc score was selected as the best-fit model to infer cline widths and centers along with a 95% confidence interval (CI). The stability and convergence of the cline parameters of the best-fit model were assessed by visualizing MCMC traces. We plotted the maximum-likelihood clines and 95% credible cline region for the best-fit model.

**Introgression**

We assigned individuals in each contact zone to hybrid classes to estimate whether gene flow is an ongoing or historic admixture. We temporarily designated individuals with $q$-values > 0.98 for $K = 2$ in the Structure analysis of sub-datasets I, II, and III as parental individuals for each cluster following *Scordato et al. (2017)*. We identified ancestry-informative markers by calculating AMOVA $F_{ST}$ for SNPs between pairs of parental clusters using the stacks program on the sub-datasets, setting three populations (two parental and one admixed population) and -$r$ = 0.80 and -$p$ = 3. The diagnostic loci, $F_{ST} = 1$, were selected as ancestry-informative markers segregating between each pair of parental clusters.

We used the R package INTROGRESS version 1.2.3 (*Gompert & Buerkle, 2010*) to calculate the maximum-likelihood estimates of the hybrid index for each individual and the average heterozygosity of each individual across informative loci. We compared genomic hybrid indices with heterozygosity to identify the individual hybrid classes. Pure individuals were defined by a hybrid index of 0 or 1 because only loci fixed in parental individuals with $F_{ST} = 1$ were used. First-generation hybrids (F1) have an expected hybrid index of 0.5 and heterozygosity of 1.0. We regarded individuals with intermediate hybrid indices (>0.25 and <0.75) and high heterozygosity (≥0.5) as recent-generation hybrids, those with intermediate hybrid indices (>0.25 and <0.75) and low heterozygosity (<0.5) as

later-generation hybrids, and those with low hybrid indices (≤0.25 or ≥0.75) as backcrossed to one or the other parental type according to previous studies (*Milne & Abbott, 2008*; *Scordato et al., 2017*; *Slager et al., 2020*).

### Estimation of migration rates

Recent migration rates between parental and hybrid populations were calculated using the Bayesian inference approach by BayesAss3-SNPs v 1.1 (*Wilson & Rannala, 2003*; *Mussmann et al., 2019*). Using sub-datasets I, II, and III after applying for the stacks program with each setting for three populations (two parental and one admixed population) and $-r = 0.80$ and $-p = 3$, BA3-SNPs -autotune v2.1.2 (*Mussmann et al., 2019*) was performed with the default parameters to find mixing parameters for BA3-SNPs. BayesAss3-SNPs was conducted with 10 million generations sampling every 100 generations using predefined mixing parameters. The first 1 million generations were discarded as a burn-in, and chain convergence was assessed in Tracer v 1.7.2 (*Rambaut et al., 2018*).

All analyses by R were conducted in R studio version 2022.07.2.576 (*RStudio Team, 2022*) using R version 4.2.2 (*R Core Team, 2022*).

## RESULTS

### Analyses of MIG-seq data

A total of 46,889,160 clean reads in 168 samples passed quality filtering, with the average percentage of reads that passed filtering for each sample being 77.6%. Among them, 17,644,888 reads were successfully mapped to the reference genome of *B. gargarizans* in the reference-mapping approach with an average mapping quality of 27.2%.

### Genetic structure and phylogeny

A total of 839 variants were identified in dataset I of 157 samples of *B. japonicus* and *B. torrenticola*.

We retained all information (157 PCs) for the initial DAPC on dataset I. After running the initial steps, the first 21 PCs were retained following the result of the *optim.a.score* function (Fig. S1A). The BIC plot in DAPC displayed the lowest value at $K = 4$ and 5 (Fig. S1B), and both clearly identified three clusters corresponding to *B. j. formosus*, *B. j. japonicus*, and *B. torrenticola*. The results of $K = 4$ identified two subclusters within *B. j. japonicus*. In addition, two subclusters within *B. j. formosus* were recognized for $K = 5$. However, these defined subclusters within *B. j. japonicus* and *B. j. formosus* had markedly overlapping plots between subclusters (Fig. 1A).

A total of 570 variants were identified in dataset I-2 of 131 samples. A Structure analysis of dataset I-2 supported two peaks for the $\Delta K$ estimation, $K = 2$ and 5 (Fig. S2A), and the number of $K$ estimated from MedMeaK, MaxMeaK, MedMedK, and MaxMedK values was 5 (Fig. S2B).

Therefore, $K = 5$ may be the valid cluster number in our results, leading to a similar grouping pattern to the DAPC (Fig. 1B). The five genetic clusters identified by DAPC and Structure analyses corresponded to northern *B. j. formosus* (NF), southern *B. j. formosus*
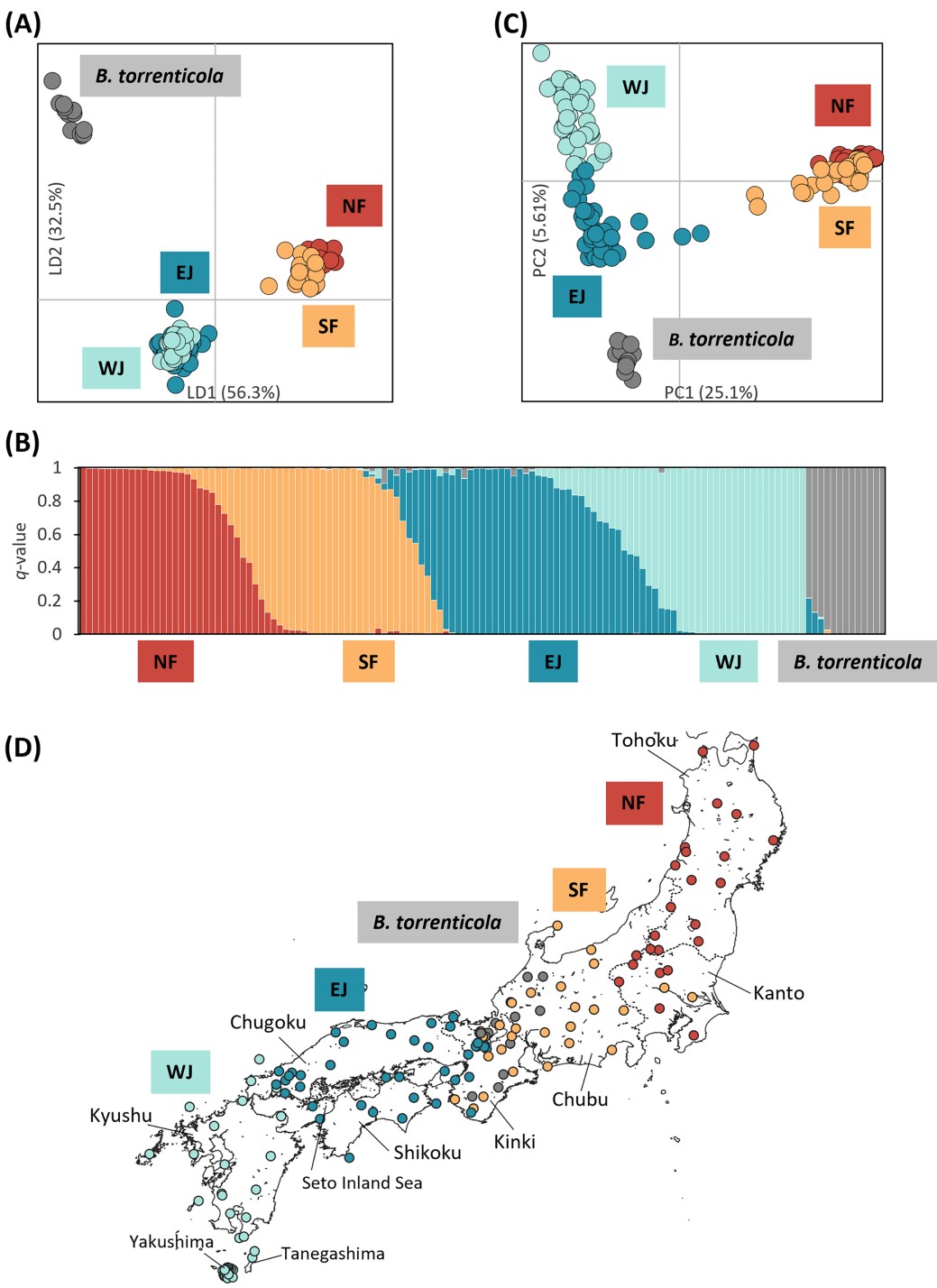

**Figure 1 Population structure using (A) DAPC, (B) Structure, and (C) PCA based on SNPs datasets, dataset I for DAPC and PCA, and dataset I–2 for Structure; (D) the distribution map of individuals colored by the cluster assignments by DAPC.** The four different genetic clusters, northern *Bufo japonicus formosus* (NF), southern *B. j. formosus* (SF), eastern *B. j. japonicus* (EJ), western *B. j. japonicus* (WJ), are displayed with *B. torrenticola*. (A) DAPC plot shows the best fit for *K* = 5 clusters. The axes represent the first two linear discriminants (LD), and the dots represent individuals colored by their groups in DAPC. (D) The distribution map of individuals colored by the cluster assignments by DAPC. The map was created by QGIS 3.28 (https://qgis.org). The administrative areas dataset was

**Figure 1** (continued)
obtained from the GADM database (www.gadm.org, version 3.4) and the inland water dataset from the Digital Chart of the World available at the DIVA-GIS online resource (www.diva-gis.org). (B) Structure bar plots show individual ancestry to the five clusters ($K = 5$). (C) PC1 and PC2 are plotted. Each dot corresponds to an individual colored according to their genetic cluster found in DAPC. The first axis distinguishes *B. j. formosus* and *B. j. japonicus*, and the second axis distinguishes *B. japonicus* and *B. torrenticola* and reflects intraspecific structure within *B. japonicus*.

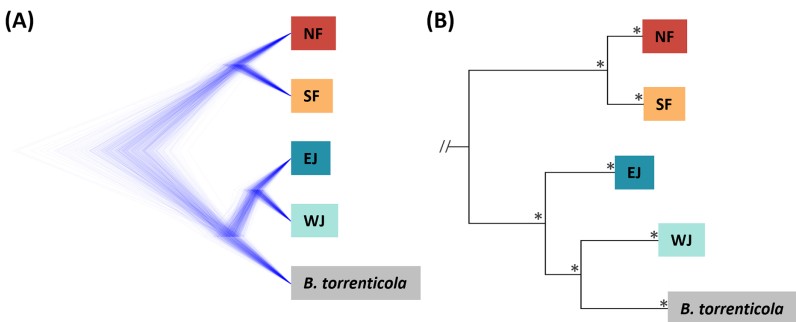

**Figure 2** (A) Densitree diagram representing the species tree obtained from SNAPP using SNPs. (B) The phylogenetic tree using mitochondrial cytochrome *b* sequences. (A) All nodes were supported by posterior probabilities of 1.0. (B) Asterisks on each node indicate bootstrap supports are more than 85%.

(SF), eastern *B. j. japonicus* (EJ), western *B. j. japonicus* (WJ), and *B. torrenticola*. Cluster assignments for individuals by DAPC are shown in Figs. 1A, 2 and Table S1.

The Structure bar plot revealed that *B. torrenticola* has rare admixtures with *B. japonicus*, three samples of *B. torrenticola* had $q$-values from 0.85 to 0.9, and one sample (Sample ID: ALC8; Table S1) of *B. j. formosus* had a $q$-value of 0.09 admixed with *B. torrenticola*. These samples appeared to be hybrid individuals based on the $q$-value threshold following *Vähä & Primmer (2006)*. Therefore, the admixed sample of *B. j. formosus* was excluded from the subsequent analysis of *B. japonicus* (datasets II, II-2, III, and III-2 and all sub-datasets).

Structure assignments also revealed hybridization between each adjacent cluster of *B. japonicus* (Fig. 1B). The admixture proportion assignment for each cluster of *B. japonicus* changed in steps. High levels of continuous admixtures were indicated across the geographic transition between NF and SF and between EJ and WJ. In contrast, hybrid individuals were limited to the boundary between SF and EJ.

The first PC axis explained 25.1% of the genomic covariance in PCA. It separated the two subspecies, *B. j. formosus* and *B. j. japonicus* (Fig. 1C). By the second PC axis, *B. torrenticola* had clearly split from *B. japonicus*. In addition, the second axis separated two continuous clusters within *B. j. japonicus*.

Based on SNAPP (290 SNPs), nuclear phylogeny confirmed deep splits between the five main clades (Fig. 2A). Mitochondrial cytochrome *b* phylogeny (1,071 bp) recovered the splits of the main clades confirmed in *Fukutani et al. (2022*; Fig. 2B*)*.

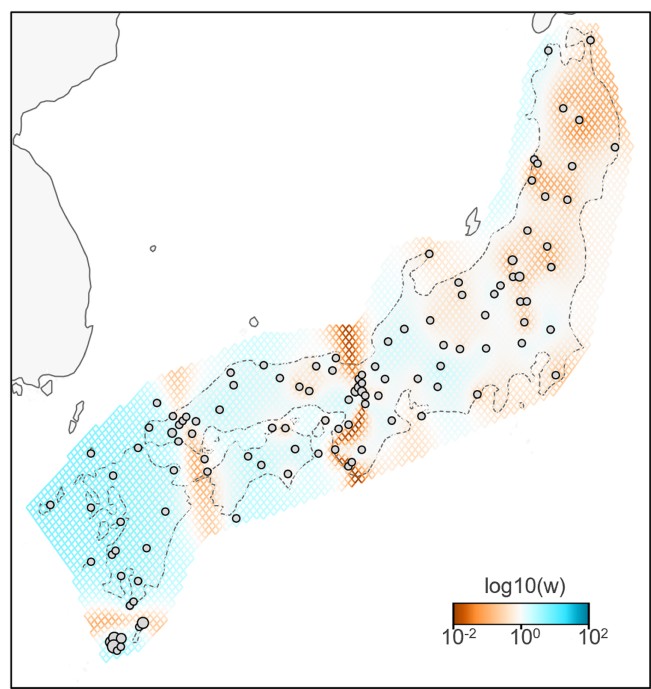

**Figure 3 Effective migration rates for the lowest cross-validation lambda estimated by FEEMS (Fast Estimation of Effective Migration Surfaces) using dataset II.** The figure shows the fitted parameters in the log scale, with lower effective migration shown in orange and higher effective migration shown in blue. Dots represent individuals.

## Artificially introduced population

A total of 718 variants were identified in the 128 samples of dataset III-2, *B. japonicus*, including the 11 samples from the artificially introduced populations in Hokkaido, Izu Islands, and the Kanto region. Two individuals in Hokkaido (Asahikawa and Hakodate) had an admixture, mainly two clusters of NF and SF, similar to those in Niijima and Kouzushima (Fig. S3). Individuals in Tokyo and Kanagawa prefectures had four admixed clusters of NF, SF, EJ, and WJ. The individual in Oshima had three clusters of SF, EJ, and WJ, and one in Hachijojima had clusters of SF and EJ.

## Effective estimates of migration surfaces

A total of 783 variants were identified in dataset II, 143 samples of *B. j. japonicus* and *B. j. formosus*. The estimated effective migration rates confirmed low migration rates between *B. j. formosus* and *B. j. japonicus* despite the absence of any geographic barrier that limits gene flow between subspecies (Fig. 3). Among *B. j. japonicus*, low migration rates were detected between Chugoku and Shikoku *vs* Kyushu, and Kyushu *vs* Yakushima, which appeared to be due to the presence of straits. In contrast, high migration rates were detected within them. On the other hand, among *B. j. formosus*, low migration rates were widely identified from Tohoku to Chubu, likely due to fewer interactions between regions than among *B. j. japonicus*.

 

## Hybrid zone analyses

Each sub-dataset consisted of cluster pairs, sub-dataset I (NF–SF) of 47 samples, sub-dataset II (SF–EJ) of 47 samples, sub-dataset III (EJ–WJ) of 59 samples, and sub-dataset III-2 (EJ–WJ excluding samples in Shikoku and Seto Inland Sea) of 48 samples. The geographic distribution of each cluster detected by *tess3r* on each sub-dataset ($K = 2$) did not significantly differ from that of Structure analyses by SNP data. The baselines across the three contact zones are shown in Fig. 4. Regarding the SNP data of sub-dataset SF–EJ and the mtDNA data of EJ–WJ, the best-supported model in *hzar* with the lowest AICc was that in which scaling was fixed to the minimum value of 0 and maximum value of 1, and no exponential tails were desired. In sub-datasets NF–SF and EJ–WJ and the mtDNA data of SF–EJ and NF–SF, the model selected was that in which scaling was fixed to the minimum and maximum observed mean values, and no exponential tails were desired.

Based on SNP data, the cline width decreased from NF–SF 170 (CI [82–362]) km to EJ–WJ 162 (CI [63–330]) km and SF–EJ 29 (CI [24–76]) km (Fig. 5). The estimated centers based on SNP data as the distance from the baseline were 0.6 (CI [−9.5 to 12]) km for SF–EJ, 5.4 (CI [−42 to 58]) km for NF–SF, and 7.0 (CI [−40 to 56]) km for EJ–WJ.

Based on mtDNA data, the cline width decreased from NF–SF 86 (CI [35–223]) km to EJ–WJ 75 (CI [31–212]) km and SF–EJ 39 (CI [18–106]) km (Fig. 5). The estimated centers based on mtDNA data as distances from the baseline were 0.3 (CI [−12 to 15]) km for SF–EJ, 23 (CI [−12 to 64]) km for NF–SF, and −33 (CI [−75 to −6.2]) km for EJ–WJ.

In addition, in the sub-dataset EJ–WJ excluding samples in Shikoku and Seto Inland Sea (48 samples), the model selected for SNP and mtDNA data was that in which scaling was fixed to the minimum and maximum observed mean values, and no exponential tails were desired. Based on SNP data, the width was 99 (CI [33–301]) km and the distance from the baseline was −1.2 (CI −38 to 53) km. Based on mtDNA data, the width was 79 (CI [32–245]) km and the distance from the baseline was −33 (CI [−76 to −2.0]) km (Fig. 5).

## Introgression

We identified loci that were informative for assigning hybrid classes for each sub-dataset. There were 40 loci with $F_{ST} > 1.0$ between parental SF and EJ, and six loci for the NF and SF pair and EJ and WJ pair. Comparisons of individual hybrid indices and average heterozygosity using these differentiated loci revealed that none of the pairs contained F1 individuals (Fig. 6). Recent-generation hybrids with high heterozygosity were detected in the NF–SF contact zone only, confirming ongoing gene flow. Later-generation hybrids were detected in all contact zones, and hybrid individuals with intermediate hybrid index values and heterozygosity of zero were identified in NF–SF and EJ–WJ contact zones, suggesting that old-origin hybrids survived. Backcrossed individuals with both parental populations were identified in the NF–SF and SF–EJ contact zones, while those with one parental population were detected in the EJ–WJ contact zone.

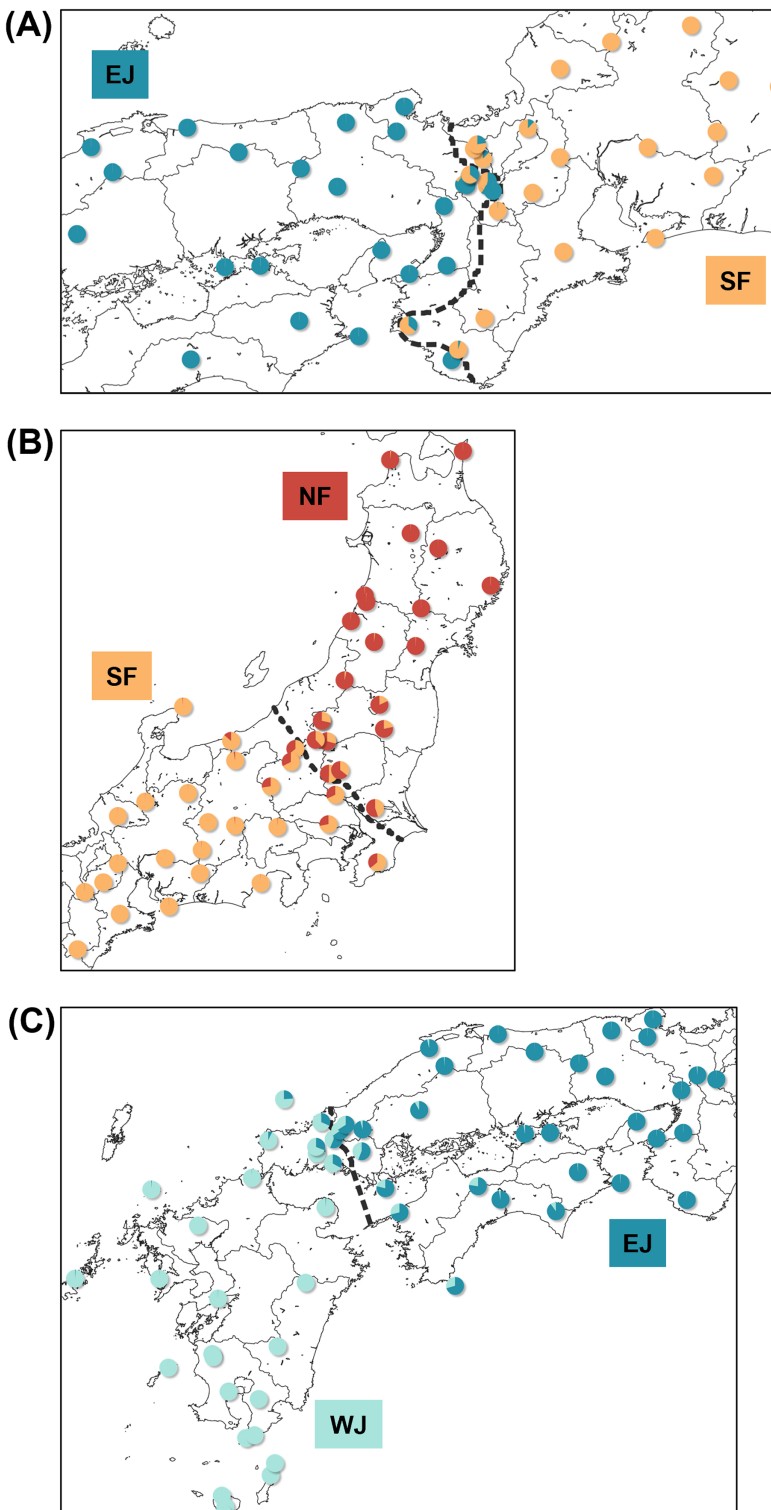

**Figure 4 Maps showing sampling localities with pie charts for three different contact zone of sub-datasets, (A) SF–EJ, (B) NF–SF, and (C) EJ–WJ.** Pie charts show the *q*-values inferred by the Structure program for each individual. The dotted lines indicate the baselines used for *hzar*.

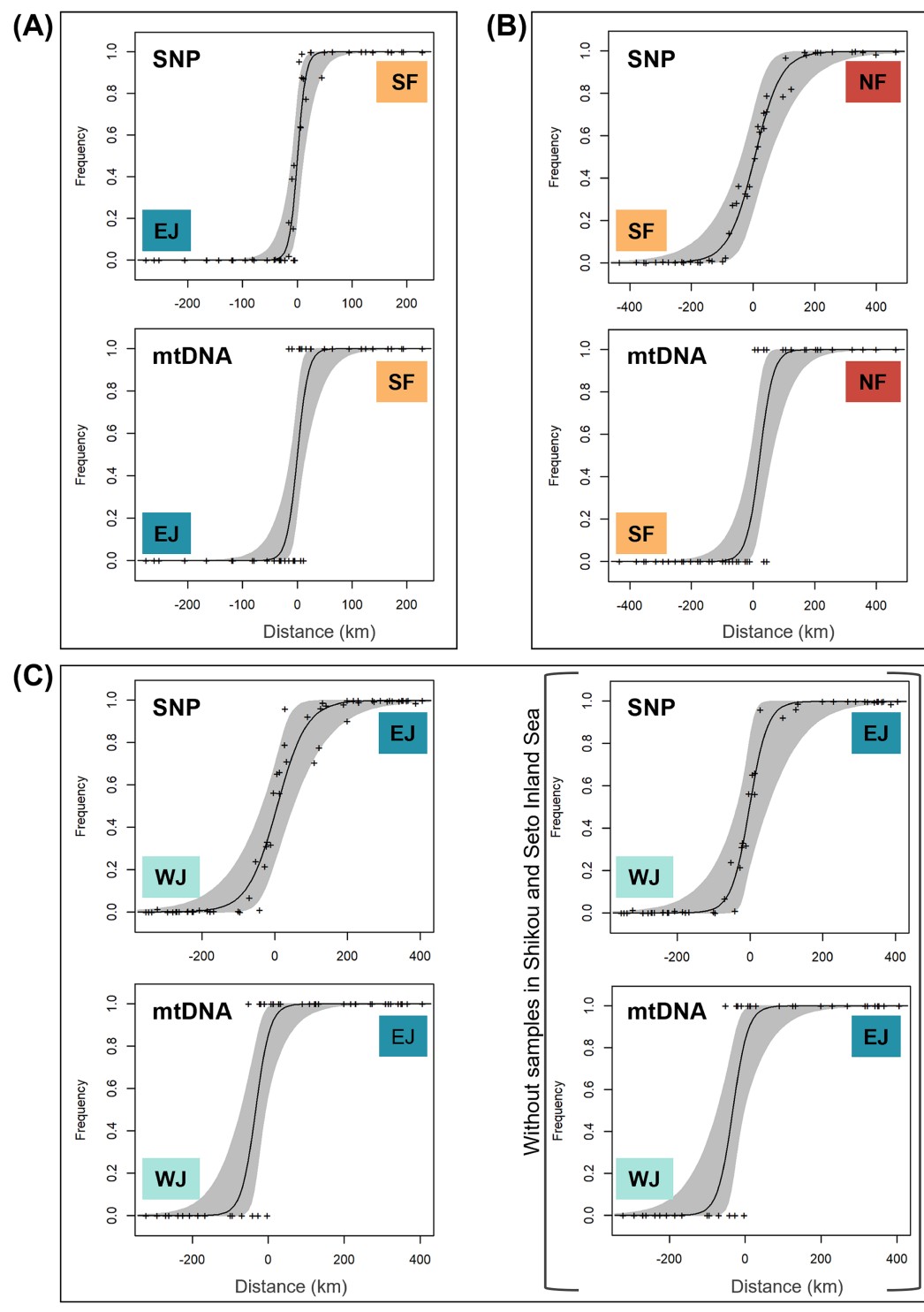

**Figure 5 The maximum-likelihood clines fitted on nuclear genomic average ancestry and mitochondrial allele frequencies along three different transects of sub-datasets, (A) SF–EJ, (B) NF–SF, and (C) EJ–WJ.** The grey areas show the 95% credible cline region. The x-axis represents distances (km) from the baselines shown in Fig. 4. Crosses indicate the observed values for individuals.

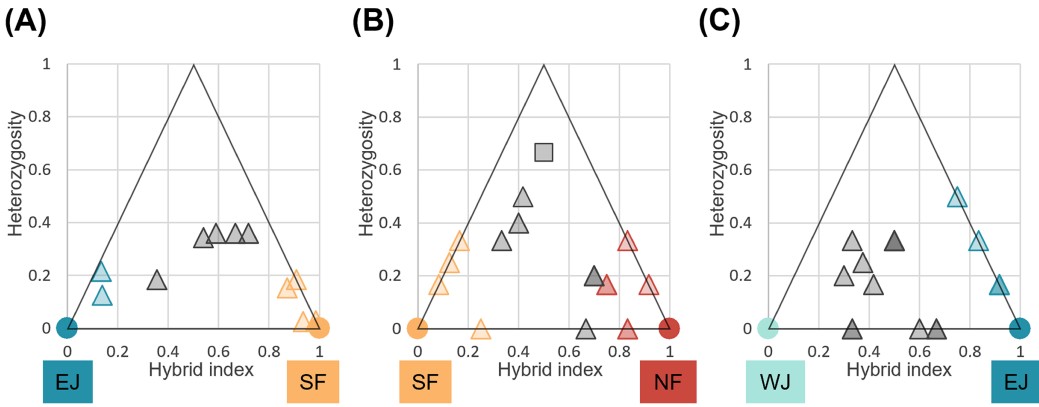

**Figure 6 Triangle plots of the hybrid index *versus* heterozygosity of individuals based on selected ancestry-informative SNP markers (*Fst* = 1) for sub-datasets, (A) SF–EJ, (B) NF–SF, and (C) EJ–WJ.** Individual with intermediate hybrid indices (>0.25 and <0.75) and high heterozygosity (≥0.5) was considered as recent-generation hybrid (a gray square), and those with intermediate hybrid indices (>0.25 and <0.75) and low heterozygosity (<0.5) as later-generation hybrids (gray triangles). Those with low hybrid indices (≤0.25 or ≥0.75) were considered as backcross to one or the other parental type (triangles colored by parental assignments). Each colored circle indicates the pure individuals.

## BayesAss directional migration

The mixing parameters for migration rates (-*m*), allele frequencies (-*a*), and inbreeding coefficients (-*f*) were selected using BA3-SNPs-autotune for each sub-dataset: sub-dataset NF–SF, -*m* = 0.2125, -*a* = 0.55, -*f* = 0.1; sub-dataset SF–EJ, -*m* = 0.2125, -*a* = 0.55, -*f* = 0.1281; sub-dataset EJ–WJ, -*m* = 0.1563, -*a* = 0.325, -*f* = 0.1.

All estimated migration rates between populations are shown in Table 1. In the sub-dataset NF–SF, the self-recruitment estimate of the parental population of SF was high at >95%, while those of the parental population of NF and the hybrid population were slightly lower (90–95%). Northward migration rates through the hybrid zone, from the parental SF to the hybrid (5.9%) and from the hybrid to the parental NF (3.6%), were higher than migration rates in the opposite direction, from the parental NF to the hybrid (1.5%) and from the hybrid to the parental SF (1.7%).

In the sub-dataset SF–EJ, self-recruitments within both parental populations were estimated to be high at >95%. In contrast, the hybrid population had low self-recruitment rates at 76.2%. Correspondingly, outward migration rates from the hybrid population into parental populations were low (2.0% to parental SF and 1.3% to parental EJ efflux). In contrast, migration rates into hybrid populations were high (16.7% from parental SF and 7.1% from parental EJ influx).

In the sub-dataset EJ–WJ, the self-recruitment of both parental populations was high at >95%, while that of the hybrid population was intermediate at 80.1%. The estimations of migration rates from hybrids into both parental populations were low (1.3% to parental EJ and 1.4% to parental WJ efflux). In contrast, migration rates into hybrid populations were high (7.3% from parental EJ and 12.6% from parental WJ influx). The migration rates among each parental population were estimated to be very low, ranging between 1.3 and 2.6%.

**Table 1 Estimates of migrants from BayesAss3-SNPs analyses between population clusters, (A) NF–SF, (B) SF–EJ, and (C) EJ–WJ.** The row headers represent the populations into where the individuals migrated, and the column headers represent the populations from where the migrant derived. Standard deviations of the values are given in parentheses.

| | | | Migration from | | |
|---|---|---|---|---|---|
| (A) | | | Parental NF | Hybrid | Parental SF |
| | Migration into | parental NF | 0.9407 (0.0349) | 0.0355 (0.0290) | 0.0238 (0.0221) |
| | | Hybrid | 0.0152 (0.0145) | 0.9262 (0.0293) | 0.0586 (0.0269) |
| | | Parental SF | 0.0167 (0.0158) | 0.0167 (0.0158) | 0.9667 (0.0218) |
| (B) | | | Parental SF | Hybrid | Parental EJ |
| | Migration into | Parental SF | 0.9607 (0.0253) | 0.0196 (0.0185) | 0.0196 (0.0185) |
| | | Hybrid | 0.1667 (0.0431) | 0.7619 (0.0389) | 0.0714 (0.0353) |
| | | Parental EJ | 0.0133 (0.0128) | 0.0133 (0.0128) | 0.9733 (0.0178) |
| (C) | | | Parental EJ | Hybrid | Parental WJ |
| | Migration into | Parental EJ | 0.9683 (0.0209) | 0.0133 (0.0151) | 0.0159 (0.0152) |
| | | Hybrid | 0.0725 (0.0280) | 0.8013 (0.0340) | 0.1262 (0.0337) |
| | | Parental WJ | 0.0139 (0.0133) | 0.0139 (0.0133) | 0.9722 (0.0184) |

## DISCUSSION

### Genetic clustering and phylogeny

Previous studies reported that Japanese toads diverged into six mitochondrial lineages from the late Miocene to the middle Pleistocene (*Igawa et al., 2006*; *Fukutani et al., 2022*). The two subspecies, *B. j. japonicus* and *B. j. formosus*, were recommended for elevation to the species level given their Miocene split. However, the findings of these studies were insufficient for the taxonomic conclusion because they were based solely on mitochondrial analyses (*Dufresnes & Litvinchuk, 2021*). Given the contact between the distribution zones of the two subspecies (*Fukutani et al., 2022*) and the possible presence of a hybrid zone between them (*Miura, 1995*), identifying the status of the zone is necessary for the study of the taxonomic status of Japanese toads because we followed the integrative species concept that considers phylogeny and reproductive isolation.

We used SNP markers of samples covering virtually the complete distribution ranges of *B. j. japonicus*, *B. j. formosus*, and *B. torrenticola* and presented the clustering and phylogenetic relationship between the identified clusters. We then showed the results of a fine-scale analysis of gene flow across the secondary contact zones of *B. j. formosus* and *B. j. japonicus*.

The consensus across independent methods suggested that $K = 5$ most accurately described the population structure of *B. japonicus* and *B. torrenticola*. This SNP clustering was roughly concordant with the five main mitochondrial clades in *Fukutani et al. (2022)*, except for the lesser diverged mitochondrial clade in the Tohoku region (NF). Based on SNP data, phylogeny confirmed the splits between the five main clades. However, the topology was discordant with the mitochondrial phylogenetic topology for the clades in western Japan (Fig. 2). The SNP phylogenetic tree showed EJ and WJ as sister clades and

supported the monophyly of *B. j. japonicus*. However, in the mitochondrial phylogenetic tree, *B. j. japonicus* was paraphyletic because *B. torrenticola* and WJ were identified as sister clades with a high node support (Fig. 2). One explanation is that *B. torrenticola* and the ancestor of EJ and WJ may all simultaneously diverge. Alternatively, discordance may stem from ancestral mitochondrial introgression between *B. torrenticola* and WJ after they diverged. These hypotheses need to be tested explicitly in future phylogenetic studies.

### The hybrid zone between *B. j. japonicus* and *B. j. formosus*

We found that mitochondrial and SNP marker cline positions and shapes varied for the three contact zones between four clusters of *B. j. japonicus* and *B. j. formosus* and showed different patterns of gene flow.

The hybrid zone between *B. j. japonicus* and *B. j. formosus* showed a sharp genetic transition, with concordant and coinciding clines between mtDNA and SNP (Fig. 5B). The cline width depended, in part, on whether the hybrid zone was structured primarily by selection or by a neutral process (*Mallet et al., 1990*). The cline width without any form of selection may be calculated using the following diffusion approximation from *Barton & Gale (1993)*: $w = 2.51\sigma\sqrt{T}$, where *w*, cline width, *T*, number of generations since secondary contact, and *σ*, average lifetime dispersal. While the lifetime dispersal distance for *B. japonicus* currently remains unknown, the maximum dispersal distance recorded for native *B. j. formosus* between the breeding pond and the summer home range is 0.26 km and the generation time is 3 years (*Kusano, Maruyama & Kaneko, 1995*). At a dispersal of 0.78 km per generation, cline width exceeds the 29.4 km width of the hybrid zone in *ca.* 677 years of unrestricted diffusion. Based on their paleo distribution, these toads came into contact with expansion after the last glacial period at the latest (*Fukutani et al., 2022*). Therefore, contact between the subspecies is arguably markedly older than 677 years. The cline width may have been kept narrow over a long time despite the absence of geographic barriers to dispersal, presumably through selection against hybrids, suggesting that the two subspecies formed a tension zone (*Key, 1968*) in the Kinki region. In addition, all hybrid individuals were classified by INTROGRESS as layer-generation hybrids or backcrosses, suggesting the relatively ancient origin of their contact.

### The hybrid zone within *B. j. japonicus*

Based on the refugia distributions proposed by *Fukutani et al. (2022)*, the mitochondrial boundary of EJ and WJ may have been maintained at the western edge of the Chugoku region from the last glacial period to the present. Therefore, EJ and WJ likely shared refugia during the glacial period, resulting in admixture. Admixed individuals may have spread to the Shikoku region and surrounding islands through the Seto Inland Sea, which covered a terrestrial and freshwater environment due to the lower sea level during the glacial period until 13,000 years ago between the western part of Chugoku and Shikoku regions (*Yashima, 1994*).

While the strait between the Chugoku and Kyushu regions formed 8,000 years ago (*Yashima, 1994*), which was later than that between the Chugoku and Shikoku regions, admixed individuals were identified in the Shikoku region, but not in the Kyushu region,

suggesting asymmetric introgression. Furthermore, this asymmetric introgression may have resulted in discordance in mtDNA and nuclear cline positions between EJ and WJ (Fig. 5C), where the mitochondrial cline center shifted approximately 40 km west from the nuclear cline center, with partially overlapping CI. The incongruity of clines inferred from different sets of molecular markers is a common phenomenon of terrestrial vertebrate hybrid zones, including amphibians (*e.g.*, *Dufresnes et al., 2014*; *Arntzen et al., 2017*; *Sequeira et al., 2020*). Prezygotic or postzygotic effects may explain the discordance in mtDNA and nuclear cline position. Sex-biased asymmetries (*Toews & Brelsford, 2012*) and an environmental gradient acting on mtDNA (*Cheviron & Brumfield, 2009*) as prezygotic effects and Haldane's rule (*Haldane, 1922*; *Orr, 1997*) and Dobzhansky–Muller incompatibilities (*Dobzhansky, 1936*; *Muller, 1942*) as postzygotic effects may have produced discordance in mtDNA and nuclear clines. Future field and genomic studies are needed to test these hypotheses and identify the factors that caused admixed individuals to spread mainly to the east of the mtDNA boundary at the time of secondary contact during the glacial period.

Regarding the width of the cline, including Shikoku, 20,490 years are needed to reach 161.8 km using the above formula, and the width of the cline, not including Shikoku, is 88.6 km which requires 6,144 years to reach, suggesting that selection may not act specifically in the Shikoku region. Furthermore, the range of present suitable habitats for EJ and WJ in *Fukutani et al. (2022)* was consistent with the actual distribution boundaries within the Chugoku region, indicating exogenous environmental factors. However, *Matsui (1984)* did not identify morphological differences between EJ and WJ. Moreover, the distribution of hybrid individuals in the Shikoku region suggests that EJ and WJ are the same species, despite the different degrees of admixture on the transect in the Chugoku and Shikoku regions.

The toad population in Yakushima was once considered to be a different subspecies of *B. japonicus* (as *vulgaris Okada, 1928*), but is now recognized as the same species based on morphology (*Matsui, 1984*). Based on mitochondrial phylogeny in a previous study (*Fukutani et al., 2022*), morphologically defined groups were not monophyletic and did not form a single cluster in this study. There may have been interbreeding between the Kyushu, Yakushima, and Tanegashima populations when the straits between Yakushima, Tanegashima and Kagoshima were terrestrial during the glacial period (*Ikehara, 1992*). Geographic isolation after the last glacial period may have led to the deviation from isolation by distance (Fig. 2).

## The hybrid zone within *B. j. formosus*

We identified the hybrid zone between NF and SF as the widest in the present study (Fig. 5A), which was an expected result because of their recent evolutionary histories (*Fukutani et al., 2022*). Widespread gene flow and recent hybridization indicate the absence of endogenous reproductive barriers between NF and SF. Furthermore, the mtDNA and SNP clines between NF and SF had an almost concordant center (Fig. 5A), suggesting the absence of selection (*Toews & Brelsford, 2012*). In contrast, the SNP cline was wider than the mitochondrial cline across the transition between NF and SF due to the

lower effective population size of mitochondrial DNA than nuclear markers (*Toews & Brelsford, 2012*).

The time needed to reach the 170-km width of the SNP cline between NF and SF without selection was calculated to be 22,619 years, suggesting a prominent role for neutral processes. According to our previously predicted distributions during the glacial period, NF and SF may have shared their refugia around the southern Tohoku to northern Kanto regions (*Fukutani et al., 2022*). The expansion of distribution after the last glacial period may have led to widespread hybridization. An expansive hybrid zone consisting of late-generation hybrids and backcrosses is consistent overall with a prolonged period of neutral expansion. Although we did not find any asymmetry for the hybrid class assignment in the triangle plots (Fig. 6), the results obtained on the direction of migration were predominantly from SF to NF through the hybrid population (Table 1), indicating that this hybrid zone will lead to the formation of a hybrid swarm in the future.

## Taxonomic revision of *B. japonicus*

Based on the above discussion, we reviewed the taxonomy of *B. japonicus*. SNP clustering based on DAPC supported four cluster numbers for *B. japonicus*, and nuclear phylogeny according to SNAPP confirmed deep splits between the five main clades.

However, based on PCA, these defined subclusters had markedly overlapping plots between NF and SF and between EJ and WJ. Additionally, hybrid zone analyses between NF and SF and between EJ and WJ indicated weak or no selection against hybrids that was insufficient for them to be regarded as different species.

In contrast, at the hybrid zone between *B. j. japonicus* and *B. j. formosus*, there was sufficient selection against hybrids for them to be regarded as different species. Hybridization persisted over time as parentals moved into the hybrid zone (Table 1). In contrast, introgression was limited by negative selection against hybrids (Table 1), allowing species to maintain their genetic distinctiveness (*Barton & Hewitt, 1985*). These results call for a taxonomic revision of *B. j. japonicus* and *B. j. formosus*. Therefore, we consider the eastern Japanese common toad *B. formosus* as a distinct species as originally described (*Boulenger, 1883*) and not a subspecies of the western Japanese common toad *B. japonicus* as currently considered (*e.g.*, *Matsui & Maeda, 2018*).

We validated the two Japanese common toads, the western Japanese toad *Bufo japonicus* Temminck and Schlegel, 1838 (type locality: Japan (for discussion, see *Matsui, 1984*)) distributed in south-western Japan, and the eastern Japanese toad *Bufo formosus Boulenger, 1883* (type locality: Yokohama, Japan) distributed in north-eastern Japan.

Morphometric variation analyses of these two species were conducted by *Matsui (1984)*. However intermediate forms were not detected in the Kinki region, and the morphological boundary extended more westerly to the Chugoku region (*Matsui, 1984*). The discordant patterns in morphological and genetic markers warrant further study.

Speciation with gene flow is common in anurans (*Dufresnes et al., 2021*). For example, a previous study on two European *Bufo* species, *B. bufo* and *B. spinosus*, which diverged in the Late Miocene, showed limited gene flow across a narrow hybrid zone (width of approximately 30 km) in the northwest of France even with the absence of barriers to

dispersal (*Arntzen et al., 2016*). Despite the presence of a hybrid zone for *B. formosus* and *B. japonicus*, the identity of the parental species is distinctive and appears to have been unaffected. These two species could be considered to remain in partial reproductive isolation over a long period (cf. *Servedio & Hermisson, 2020*). Cline coupling may have progressed further towards reproductive isolation after secondary contact, and it may still be ongoing throughout the hybrid zone (*Harrison & Larson, 2014*; *Butlin & Smadja, 2018*).

We also found that the geographic location of the hybrid zone between the two species appeared to be independent of the environment. Ecological niche modeling in *Fukutani et al. (2022)* showed that environmental conditions were suitable for both species across the hybrid zone identified in this study, suggesting that environment-associated selection may not act directly to keep the hybrid zone. Many anuran speciation processes are initiated through the gradual accumulation of multiple barrier loci scattered across the genome, which reduces hybrid fitness by intrinsic postzygotic isolation (*Dufresnes et al., 2021*). Similarly, for *B. formosus* and *B. japonicus*, many genomic regions may represent local barriers to gene flow. We will attempt to elucidate the genomic mechanism that induces speciation in future studies.

## CONCLUSION

In summary, we presented three hybrid zones with different cline shapes. Populations with greater divergence had a sharper hybrid zone cline. These results on Japanese toads are consistent with other findings on anuran species (*e.g.*, *Dufresnes et al., 2018*, *2020c*). They are also very applicable to the most deeply diverged populations, *B. japonicus* and *B. formosus*, which had a sharp cline, suggesting the presence of strong selection (*Mallet et al., 1990*). Our results will contribute to resolving taxonomic confusion in Japanese toads.

## ACKNOWLEDGEMENTS

We acknowledge K. Eto, I. Fukuyama, R. Fukuyama, S. Ikeda, K. Kimura, Y. Misawa, Y. Miyagata, T. Shimada, T. Sugahara, T. Sugihara, Y. Tahara, S. Tanabe, A. Tominaga, N. Yoshikawa, and many more collaborators for collecting samples. We thank N. Yoshikawa and Y. Fuke for helping to conduct MIG-seq and analyses. We also thank our laboratory members for helping with specimen processing and molecular experiments. Finally, we thank the reviewers for their valuable comments.

### Funding

This work was supported by the JSPS KAKENHI Grant (JP21J15839), the Environment Research and Technology Development Fund (JPMEERF20204002) of the Environmental Restoration and Conservation Agency of Japan, and the Sasakawa Scientific Research Grant from the Japan Science Society. The funders had no role in study design, data collection and analysis, decision to publish, or preparation of the manuscript.

## Grant Disclosures

The following grant information was disclosed by the authors:
JSPS KAKENHI: JP21J15839.
Environment Research and Technology Development Fund: JPMEERF20204002.
Environmental Restoration and Conservation Agency of Japan.
Sasakawa Scientific Research Grant from the Japan Science Society.

## Competing Interests

The authors declare that they have no competing interests.

## Author Contributions

- Kazumi Fukutani conceived and designed the experiments, performed the experiments, analyzed the data, prepared figures and/or tables, authored or reviewed drafts of the article, and approved the final draft.
- Masafumi Matsui conceived and designed the experiments, authored or reviewed drafts of the article, and approved the final draft.
- Kanto Nishikawa conceived and designed the experiments, authored or reviewed drafts of the article, and approved the final draft.

## Animal Ethics

The following information was supplied relating to ethical approvals (*i.e.*, approving body and any reference numbers):

The Animal Experimentation Ethics Committee in the Graduate School of Human and Environmental Studies, Kyoto University provided full approval for this research (20-A-5, 20-A-7, 22-A-2).

## Data Availability

The raw sequence reads data are available at DDBJ DRA: DRA016475.

## Supplemental Information

Supplemental information for this article can be found online at http://dx.doi.org/10.7717/peerj.16302#supplemental-information.

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
