# Peer review of "Population genetic structure and hybrid zone analyses for species delimitation in the Japanese toad (Bufo japonicus)"

_PeerJ, doi:10.7717/peerj.16302_

## Round 0.1 · original submission · Major Revisions

· Academic Editor

Major Revisions

I have now received the comments from three reviewers on your manuscript. While two of them only suggested minor changes, mostly on writing and clarifying the species concept that supports the study, R1 makes more profound critiques, focusing on the framing of the paper. I tend to agree with her/him, and most of the confusing parts that are pointed out by R2 and R3 can be linked to what R1 highlights.

In the Methods, make sure the description of the bioinformatics and molecular techniques are thorough enough to allow reproducibility. I’m not sure I understood your use of DAPC and PCA to analyse the same data set. They seem redundant. Please, double check that

Reviewer 1 ·

Basic reporting

This paper seeks to evaluate the taxonomic status of several groups within the toad Bufo japonicus. The authors sampled extensively and obtained a large amount of molelar genetic data and did a very good job at that. Yet, the final outcome is somewhat unbalanced because evolutionary mechanisms remain insufficiently addressed (while the data are there) whereas the taxonomic treatment is incomplete. In detail my main concerns are as follows.

Concern 1 - While it is claimed that 'results provide insights into the role of the hybrid zone on speciation and the processes that create and maintain biodiovcersity' (line 613) this conclusion is not put into effect. The focus of the paper is on taxonomy whereas evolutionary investigations are postponed (lines 485, 540, 563).

Concern 2 - The paper suffers from an analytical overkill. What is the purpose in analyzing the data by three methods (Structure, DAPC, PCA). Similarly, what is the point of doing a HZAR analysis and produce geneflow surfaces and the estimation of migration rates ?

Concern 3 - This essentially being a taxonomic paper, some crucial information is lacking, most prominently for the newly recognized Bufo formosus. What is the type locality ? Where are the types and paratypes located ? Is a proper formal diagnosis available from Boulenger (1883) ? If not, you should provide one here, because someone else is not going to do this for you ! What is the situation for Bufo japonicus Temminck and Schlegel, 1838 ? Take the opportunity and settle these issues !
On a related point, within both species subgroups are being recognized as east verus west (within Bufo japonicus) and north versus south (within Bufo formosus). Assuming that you consider these to be subspecies (because contact zones are wise and shallow), you are in a position to name them, before someone else does, but this of course involves some effort.

Concern 4 - line 148-151. Introduced populations. Was the introduced status of these supposedly introducd populations known or suspected prior to the study ? If so, what is the relevance of this research sideline; why did you pay attention ? If not, is the introduced status perhaps an ad hoc explanation for results that do not quite fit the grand scheme of things ? Are introductions of other species known for the same region ? I see data being presented (line 372) but not being discussed.

Concern 5 - While clear and mostly unambiguous, the English is at places a bit clumsy and the manuscript could do with some revision by a fluent speaker.

Minor remarks

line 51 - MIG-seq. Do not use abbreviations in abstract

line 76 - fine to mention the taxonomic autority for the species, but be consistent and do this for all taxa mentioned, such as Bombina variegata, etc.

line 79-81 - 'is correlated' >> 'has been reported to correlate'

line 102 - ISSR, MIG-seq, explain abbreviations at first usagehttps://translate.google.com/

line 119 - I take it that 'samples' means tissue samples taken from individual toads ?

line 127 - is it Mig-seq or MIG-seq ?

line 176 - please number datasets as A, B, C or I, II, II, not i), ii), iii)

line 214 - 'populations' >> 'population groups'

line 241 - I found this section very long and not free of repetition. Condider making a denser version.

line 382 - present data without reference to the literature. The noted correspondece with mtDNA typing is to be dealt with in the Discussion.

line 403 - do the two selected models really have the same AICc scores ?

Figure 4 - do the gene flow surfaces extend to beyond where toads can be found (i.e oceans) ?

Experimental design

See above.

Validity of the findings

See above.

·

Basic reporting

This is an interesting manuscript on the evolution of a amphibian species complex that has been is rather well known, thus opens many questions on detailed points of its history.

Some notes on basic reporting:
English should be revised. I made some changes but I am no native speaker.

The subject is well presented and details on knowledge on the toad lineages from Japan explained; literature on the theoretical and specific aspects of the subject cited.

One aspect should be mentioned in the introduction as it is important for the study. The choice of methods is linked to the kind of species concept that is used. If strictly keeping an evolutionary species concept than analysis of genetic situation in hybrid zone would not be necessary. This is necessary if the biological and genetic situation in the contact zone of two taxa is used to analyse taxonomic situation. Thus it is linked to the biological species concept and those close that use relationships between taxa to analyse taxa status (Mayr 1942, 1982). There are recent proposals for a integrative species concept (de Queiroz 2007, 2020) that considers both aspects, phylogeny and reproductive isolation mechanism. I think this should be clearly mentioned in a work that uses analysis of hybrid zones for taxonomic definition of taxa.


No formal problems

Figures checked.
The outlines of Japanese islands would better be continuous. A map giving the names of the regions discussed would be useful for the reader.

Raw data checked.

Experimental design

Methodology used correspond to the research question defined. It gives detailed data on the fine scale evolution of a group of toads. Methodology is clearly presented and allows replication.

Validity of the findings

The findings are statistically well supported and conclusions are based on literature and results presented in this work.

Additional comments

A few points should be considered. They are noted on the manuscript.

I made some formal annotations in the References section. They need to be checked throughout

·

Basic reporting

Fukutani, et al. detail the history of the species complex and the difficulty in delineating these species in light of the multiple hybrid zones present within the subspecies Bufo japonicus japonicus and Bufo japonicus formosus. The authors used a wide variety of population identification and species delimitation methods to classify population structure and substructure within these subspecies and present evidence that supports their claim for elevation of these subspecies to full species. I think this manuscript contains a great deal of work which supports the author’s assertions, but can use some editing of the text and portions of the figures, which are noted in the review documents returned.

Experimental design

no comment

Validity of the findings

no comment

Additional comments

I thought this manuscript contained a large and impressive amount of work. Aside from my overall viewpoint that each whole subspecies (as opposed to individual clades within the subspecies) should be elevated (see comment within reviewed manuscript), I think that with some minor editing this may be very ready for publication.

---

## Round 0.2 · Minor Revisions

· Academic Editor

Minor Revisions

I have now receivd back the final comments from one of the reviewers from the previous round. Like her/him, I'm satisfied with the way authors addressed the critiques, but there're still a couple of minor changes that need to be addressed. See comments below. Also, try to format the abstract in the structured form as used by PeerJ.

·

Basic reporting

As the manuscript is in its second round of review, this has been accepted in the first round.

Experimental design

No critics in the first round.

Validity of the findings

see above.

Additional comments

Besides the points below, there are a few overlooked typing errors in the joined annotated manuscript file.

Line 602: Bufo japonicus Temminck and Schlegel, 1838 (type locality: Japan). The original type-locality (protonymotope of Frétey et al. 2018, Bionomina) could be limited to the knowledge we have of the voyage of the collector, Siebold, thus leading to the recognition of the ergonomytope. Siebold stayed in Nagasaki, while visiting Japan (Tominaga & Matsui 2007) which corresponds with the distribution of Bufo japanicus.

Line 515-16: I think this should be reformulated as from the phrasing it would appear that Servedio and Hermisson (2020) proposed the partial reproductive isolation of Bufo formosus and B. japonicus. Their work was on Spea, so I conclude that you argument on the reproductive isolation of Bufo species using their authority. Rephrase.

It was a pleasure to read this manuscript.

---

## Round 0.3 · accepted · Accept

· Academic Editor

Accept

Thank you for making these final adjustments to the text. I'm happy to recommend it for publication as is.